# Mechanical Behavior of Carbon-Fiber-Reinforced Polymer Composites (Towpreg) Under Various Temperature Conditions

**DOI:** 10.3390/polym17243241

**Published:** 2025-12-05

**Authors:** Yoonduck Seo, Jiming Sun, Amit Dixit, Da Hye Kim, Yuen Xia, Sung Kyu Ha

**Affiliations:** 1Department of Mechanical Engineering, Hanyang University, 222 Wangsimri-ro, Seongdong-gu, Seoul 04763, Republic of Korea; ydseo1216@hanwha.com (Y.S.); windymonde12@hanyang.ac.kr (J.S.); ab3628229@naver.com (D.H.K.); 2Hanwha Ocean, 3370, Geoje-daero, Geoje-si 53302, Republic of Korea; 3Aditya Birla Group, Aditya Birla Centre, S.K. Ahire Marg, Worli, Mumbai 400030, India; amit.dixit@adityabirla.com; 4Intelligent Manufacturing Institute, Yancheng Polytechnic College, No. 285, Jiefang South Road, Yancheng City 224000, China; 2025170763@ycpc.edu.cn

**Keywords:** Towpreg, hydrogen pressure vessels, dry winding, localized thermal control, Joule–Thomson effect, thermo-mechanical stability, mechanical behavior, static strength, fatigue life, S–N curve, fatigue limit, Basquin equation

## Abstract

As the hydrogen economy rapidly expands, carbon-fiber-reinforced polymer composites (Towpreg) have become key materials for next-generation hydrogen pressure vessels, offering superior processability, reproducibility, and storage stability compared to conventional wet-winding composites. Since hydrogen storage vessels are evaluated at three representative service temperatures (−40, 25, and 85 °C), Towpreg materials must maintain consistent mechanical performance across this range to meet certification standards. This study establishes an integrated methodology combining Towpreg panel fabrication, temperature-controlled tensile and fatigue testing, and quantitative assessment of thermo-mechanical stability using DM epoxy resin as the matrix. To address artifacts such as tab slippage at high temperatures and inefficiency at low temperatures, a “Localized Thermal Control” approach was developed. The HY-Mini Heater System enables localized heating at 85 °C, while the HY-Cooler System applies a Joule–Thomson-based Stirling cooler for efficient localized cooling at −40 °C. Quantitative evaluation showed tensile strengths of 2973.3 MPa (RT), 2767.3 MPa (HT, ~7% decrease), and 2907.7 MPa (LT, ~2% decrease). Under R = 0.1 fatigue testing, the Basquin slope (m) was 11.97 (RT), 9.98 (HT), and 10.6 (LT), while the intercept (log b ≈ 3.7) remained nearly constant. These results confirm the excellent thermo-mechanical stability of the carbon-fiber-reinforced polymer composites (Towpreg) for hydrogen tank applications.

## 1. Introduction

The global transition toward a hydrogen economy critically depends on the development of safe, lightweight, and durable storage solutions [1,2,3,4]. In this regard, the technology to store high-pressure (700 bar) hydrogen on-board is paramount for the commercialization of Hydrogen Fuel Cell Electric Vehicles (FCEVs) [2,5]. Type IV Composite Overwrapped Pressure Vessels (COPVs), which feature a polymer liner fully wrapped with a carbon-fiber–resin composite, are considered the most advanced solution, offering both weight reduction (approximately 70% compared to Type I vessel) and high storage efficiency (approximately 6 wt%) [6,7,8,9]. The primary bottleneck in the popularization of hydrogen FCEVs lies not in the fuel cell technology itself, but in the performance, durability, and cost of storage vessels [10]. A significant portion of this cost is driven by the use of expensive carbon-fiber composites [5,10,11]. Therefore, verifying and optimizing the performance of this composite system will determine the success of the entire hydrogen mobility industry [12]. Beyond hydrogen pressure vessels, fiber-reinforced polymer (FRP) composites have been widely applied in civil engineering structures, aerospace components, wind-energy systems, and automotive lightweighting. Recent studies highlight that FRPs provide high strength-to-weight ratios, corrosion resistance, and durability across diverse structural applications [13]. These broad applications underscore the importance of understanding temperature-dependent mechanical behavior of FRP materials.

Historically, Type IV vessels have been manufactured using the wet-winding filament winding process [14,15]. This method involves passing a dry fiber bundle through a resin bath immediately before winding it onto a mandrel. However, this technique suffers from critical disadvantages, including difficulty in precisely controlling the resin content (RSD ± 5~10%), high void content (1.0~3.0%), and poor reproducibility due to slow process speeds. In response to these challenges, Towpreg—a bundle of carbon fibers pre-impregnated with resin—has emerged as a primary alternative [6,16,17]. Towpreg provides high processability, excellent reproducibility, and superior storage stability [14]. The Towpreg process not only ensures consistent quality by precisely controlling resin content (RSD ± 1~2%) but also enhances load transfer from the matrix to the fibers, thereby maximizing mechanical properties [14,18]. This is not merely a process improvement but a paradigm shift, elevating the fabrication of safety-critical hydrogen vessels from the realm of “craft” to “industry”.

International certification standards for hydrogen storage vessels (e.g., KGS, ECE R134) mandate verification of the material’s structural integrity across a wide range of service temperatures. This range is typically set from −40 °C (during cold-weather parking or refueling) to +85 °C (a “hot soak” condition caused by engine heat or direct sunlight). The design life of a hydrogen vessel is determined by its fatigue life over thousands of charge–discharge cycles, and this fatigue behavior is highly sensitive to temperature [15,19]. Data obtained at room temperature (25 °C) alone cannot predict the material’s behavior [20]. At high temperatures, the polymer matrix can soften, leading to reduced stiffness and strength [4,15,21,22]. At low temperatures, the matrix can become brittle, potentially introducing new damage mechanisms such as microcracking [15,23,24,25]. The thermal behavior of the composite is governed by the temperature-dependent properties of the epoxy polymer matrix. Therefore, complete characterization of static and fatigue properties at all three temperature conditions is indispensable for material qualification.

The standard method for high- and low-temperature composite testing involves placing the entire specimen and test grips inside a large environmental chamber [26,27]. However, this approach can introduce important testing artifacts, particularly in high-temperature conditions, because the chamber heats not only the specimen’s gauge length but also the artifact load-transfer tabs and grips to 85 °C. At this temperature, the tab-bonding epoxy softens, which may lead to premature tab slippage or debonding before the composite specimen itself fails [28]. As a result, the measured response may be influenced by adhesive limitations rather than the intrinsic properties of the composite, potentially leading to an underestimation of high-temperature strength and fatigue performance. Therefore, temperature-chamber-based methods may present certain limitations when accurate high-temperature mechanical characterization is required. Furthermore, for low-temperature testing, large chambers are energy-intensive, require long stabilization times, and depend on costly consumables such as liquid nitrogen (LN_2_).

The first objective of this study is to address these methodological limitations. To this end, a new “Localized Thermal Control” testing paradigm that eliminates test-induced artifacts is developed and validated.

The second objective is to use this reliable methodology to comprehensively characterize the thermo-mechanical behavior (static and fatigue) of the carbon-fiber-reinforced polymer composite (Towpreg), which is emerging as a next-generation hydrogen vessel material.

The final objective is to analyze this reliable new data to assess the suitability of the carbon-fiber-reinforced polymer composites (Towpreg) for meeting Type IV hydrogen pressure vessel certification requirements.

## 2. Materials and Methodology

### 2.1. Constituent Materials

The materials used in this study were specifically designed for the fabrication of Type IV hydrogen pressure vessels. Hyosung H2250-24K carbon fiber (HYOSUNG ADVANCED MATERIALS, Seoul, Korea) was used as the reinforcement, and Aditya Birla’s CeTePox DM-2021-11-4 A/AM 5597/AM XP 332 C epoxy system (CTP Advanced Materials GmbH, Rüsselsheim, Germany) was used as the matrix resin. This DM epoxy system is a toughened epoxy developed specifically for Towpreg applications [29].

Key properties of the materials are summarized in Table 1 and Table 2. This material system is optimized for this application. Notably, the resin’s low initial mixing viscosity (590–600 mPa·s at 50 °C) is crucial for enabling fast and complete fiber impregnation, thereby minimizing void formation. Furthermore, the high glass transition temperature (T_g_) of 127 °C is a key design parameter, deliberately chosen to be well above the maximum service temperature of 85 °C to ensure high-temperature stability. The resin-impregnated towpreg is shown in Figure 1.

### 2.2. Manufacturing Process of Towpreg Flat Panel and Specimens

To ensure data reliability, a detailed panel fabrication process reproducible by third parties was established. The experimental flow chart is presented in Figure 2.

The process consists of two stages: dry winding and hot-press curing. First, a Towpreg layup of 2 (unidirectional 2-ply) was wound onto a flat mold using a dry winding machine. Key process parameters for reproducibility—winding tension (4 kg) and winding speed (120 mm/s)—were kept constant. The dry winding process is illustrated in Figure 3 and Figure 4:

Subsequently, a steel mold specifically designed to enable precise thickness control and resin flow management was employed. The material preparation for flat panel fabrication is shown in Figure 5. On the mold surface, a 1.2 mm-thick aluminum shim was placed for accurate thickness calibration, and a 2.0 mm-thick silicone dam was installed to prevent resin overflow. After the Towpreg was wound onto the mold, a peel ply, perforated release film, and bleeder were sequentially applied and fixed in position using high-temperature adhesive tape. The mold configuration, stack sequence, and stacking process are presented in Figure 6, Figure 7 and Figure 8.

The wound panels were cured in a hot press. The cure cycle is shown in Table 3 and Figure 9, and is scientifically based on the resin’s Technical Data Sheet (TDS) properties (T_g_, gel time). Specifically, the initial stage of holding at 110 °C (gel time 20–25 min) for 1 h is designed to allow resin flow, remove internal voids, and achieve consolidation before the main cross-linking at 120 °C.

After curing was completed, the demolding process was performed in the reverse order of the stacking sequence, as shown in Figure 10. The caul plate was first removed, followed by the sequential removal of the bleeder, perforated release film, and peel ply, resulting in the recovery of a flat composite plate.

The surface morphology of the cured flat plate was examined using optical microscopy, as shown in Figure 11. The surface in contact with the peel ply exhibited a woven texture pattern corresponding to the fabric structure of the peel ply, indicating successful surface replication during curing. In contrast, the surface in contact with the steel plate displayed clearly defined carbon fiber alignment and uniform resin impregnation without signs of resin starvation, confirming proper consolidation and impregnation quality.

The specimen fabrication process is shown in Figure 12. First, the panel is cut up to the surface that was pressed by the caul plate to obtain a clean and uniform test surface. Subsequently, tabs are machined and cut to the appropriate dimensions for gripping by the testing machine. The tabs are bonded to the specimen using a structural adhesive. To ensure sufficient adhesion strength, the contact surfaces between the tabs and the panel are abraded with sandpaper to increase surface roughness. After bonding, the panel is cut to the specified dimensions, thereby completing the preparation of the test specimens.

### 2.3. Novel Localized Thermal Control Systems

#### 2.3.1. High-Temperature (85 °C) “HY-Mini Heater System”

To resolve the ‘tab slippage’ artifact in high-temperature testing, the HY-Mini Heater System was designed and fabricated. The test methods for high-temperature conditions are presented in Figure 13. This system uses a compact heating pad and chamber (120 mm × 37 mm) to locally heat only the specimen’s gauge-length region to 85 °C, while the load-transfer tabs and machine grips remain at ambient temperature. This design thermally decouples the gauge region, where material failure occurs, from the tabs, preventing artifacts caused by adhesive softening. System reliability was validated using three thermocouples (TCs): one for control and two for verification. Validation confirmed that the temperature within the gauge region was maintained uniformly at 85 °C ± 1.5 °C, with a rapid stabilization time of less than 20 min.

#### 2.3.2. Low-Temperature (−40 °C) “HY-Cooler System”

To address the inefficiency and cost of low-temperature testing, the HY-Cooler System was developed as Figure 14. This system uses a free-piston Stirling cooler (FPSC) with helium (He) gas as a working fluid. The cooler efficiently generates −40 °C cryogenic temperatures by operating on the Stirling thermodynamic cycles.

The cold is transferred from the cooler tip to the specimen chamber via a passive thermosiphon. Furthermore, the chamber uses a dual-insulation structure (internal aerogel and external foil) to minimize external heat pressure. Validation using three TCs confirmed that the target temperature of −40 °C was reached within 60 min and maintained stable. The test methods for low-temperature conditions are presented in Figure 15.

These two systems represent a new paradigm for composite testing that is (a) more accurate by eliminating artifacts, (b) more efficient due to faster stabilization, and (c) more cost-effective by eliminating consumables like LN_2_, compared to conventional large-chamber methods. Demonstration of ±1.5 °C temperature uniformity via 3-TCs validation highlights the strengths of the proposed localized thermal control system—providing stable and confined temperature regulation along the gauge length. 

### 2.4. Mechanical Test Procedure

All mechanical tests were performed using a universal testing machine (UTM).

Static Tensile Test: Conducted according to ASTM D3039 at three temperature conditions: −40 °C, 25 °C, and 85 °C [30].

Fatigue Test: Tension-tension fatigue tests were conducted according to ASTM D3479 at the three temperature conditions [31]. The theoretical background of the fatigue test is presented in Figure 16.

A stress ratio of R = (σ_min_\σ_max_) = 0.1 was set to simulate the charge–discharge cycles of a pressure vessel, the fatigue tests were conducted at a loading frequency of 1 Hz.

Data Analysis: The collected fatigue data (S–N curve) were analyzed using the Basquin equation:

## 3. Results

### 3.1. Static Tensile Behavior Analysis

The representative stress–strain curves from the static tensile tests at all three temperature conditions showed distinct linear-elastic behavior until failure, a typical characteristic of unidirectional (2-ply) laminates in which the fibers carry the primary load. The strength evaluation results at the three temperatures conditions are presented in Figure 17.

Table 4 summarizes the main static tensile properties at each temperature condition. The most notable finding is the excellent thermo-mechanical stability of the carbon-fiber-reinforced polymer composites (Towpreg). The tensile strength at high temperature (85 °C, 2767.3 MPa) showed only a 7.0% decrease compared to room temperature (2973.3 MPa). The strength at low temperature (−40 °C, 2907.7 MPa) showed a negligible 2.2% decrease from room temperature. Consequently, the tensile strength variation was controlled within 7% across the entire 125 °C operating ranges from −40 °C to +85 °C.

Furthermore, the experimental Young’s Modulus at room temperature (141.1 GPa) showed a high 97.3% agreement with the theoretical value calculated by the Rule of Mixtures (145 GPa). This verifies that the fabrication process described in Section 2.2 produced high-quality specimens with minimal void content, confirming the reliability of the measured strength values.

### 3.2. Fatigue Life (S–N) Behavior Analysis

Fatigue test results (R = 0.1) for all three temperature conditions are presented as S–N curves (log–log scale), with the corresponding Basquin equation parameters summarized in Table 5.

Figure 18 presents the S–N curve for DM-Towpreg at −40 °C. Under the low-temperature (−40 °C) condition, the slope of the S–N curve was 10.6, which is slightly lower than that obtained at room temperature. This indicates that cryogenic exposure has no significant influence on the fatigue damage mechanism [23]. The fatigue limit was determined to be 1.36 GPa, demonstrating the high stiffness of the matrix and the absence of brittleness or microcrack formation even at −40 °C.

Figure 19 presents the S–N curve for DM-Towpreg at room temperature. The S–N curve at room temperature exhibited a smooth and monotonic decline with a high slope (m = 11.97), confirming the material’s excellent fatigue resistance. The fatigue limit was approximately 1.57 GPa, indicating a stable matrix–fiber interfacial bonding and consistent load transfer even under cyclic loading conditions.

Figure 20 presents the S–N curve for DM-Towpreg at 80 °C. The slope (m = 9.98) of the S–N curve at 85 °C, became slightly steeper than that at room temperature, suggesting that minor matrix softening led to a greater sensitivity of fatigue life to applied stress. Despite the elevated temperature, the linearity of the $log\sigma -\;logN_{f}$ relationship remained clear, and the fatigue limit (1.25 GPa) indicated that the thermal-induced degradation in performance was limited to approximately 20% compared with room temperature.

The fatigue test results at the three temperatures conditions are presented in Figure 21. Across the entire temperature range from −40 °C to +85 °C, the three S–N curves exhibited nearly identical slopes and intercepts, confirming that temperature had a negligible effect on the fatigue behavior of the carbon fiber reinforcement polymer composite (Towpreg). The convergence of the log b values (3.67–3.70) and the small variation in the m values (10 ± 1) demonstrate high reproducibility and thermo-mechanical stability of the material system within the service temperature range relevant to hydrogen-tank operation.

The most significant finding from the fatigue analysis is that the Basquin intercept (log b) remained nearly constant across all three temperature conditions: 3.70 (RT), 3.70 (HT), and 3.67 (LT), showing negligible variation. In the Basquin equation, the intercept (log b) corresponds to the material’s intrinsic fatigue strength characteristics.

## 4. Discussion

### 4.1. A New Paradigm in Composite Testing: Validation of the Localized Thermal Control Methodology

The core academic contribution of this study lies in the test methodology itself. The data presented herein is considered more reliable than that obtained using conventional chamber-based methods [33]. By fundamentally eliminating the tab slippage artifact during high-temperature (85 °C) testing [34], the measured tensile strength (2767.3 MPa) and fatigue limit (1254 MPa) accurately represent the intrinsic properties of the material rather than limitations of the test apparatus. This finding suggests that the high-temperature performance of composites reported in prior literature may have been systematically underestimated due to this tab slippage issue. Therefore, the “specimen-based localized thermal control” method proposed in this study should be regarded as a promising alternative for accurate thermo-mechanical characterization.

### 4.2. Superior Thermo-Mechanical Stability of the Carbon Fiber Reinforcement Polymer Composite (Towpreg)

This study provides a clear physical explanation for the exceptional thermo-mechanical stability exhibited by the carbon fiber reinforcement polymer composite (Towpreg) [35].

(1)High-Temperature Stability: The limited 7% decrease in tensile strength at 85 °C is primarily attributed to the high T_g_ of the DM resin (127 °C). Since the service temperature (85 °C) remains well below T_g_, the matrix maintains a robust glassy state rather than entering the transition region [36]. This prevents the typical degradation phenomena observed when T approaches T_g_, such as matrix softening, or plasticization. Regarding the fiber–matrix interfacial Strength and relaxation behavior, which are critical concerns at elevated temperatures, the stability of the fatigue limit and the constant Basquin intercept (log b) observed in this study suggest that the interfacial bonding integrity was maintained without significant relaxation-induced degradation. Furthermore, while residual thermal stresses typically induce microcracking at cryogenic temperatures, the high Tg margin at 85 °C minimizes the impact of thermal stress on the static strength.(2)Low-Temperature Stability: The suppression of strength variation to within 2% at −40 °C is equally significant, indicating the absence of low-temperature embrittlement. Many composite systems experience residual thermal stress from the mismatch in the coefficients of thermal expansion (CTE) between fibers and the matrix during cooling, often resulting in matrix microcracking and strength loss. The stable performance of the carbon-fiber-reinforced polymer composites (Towpreg) even under low-temperature conditions suggests that the matrix possesses high toughness and that fiber–matrix adhesion is strong, enabling the material to effectively resist internal stress concentrations.

### 4.3. Temperature-Dependent Fatigue Damage Mechanisms

The fatigue test results provide deeper insight into the influence of temperature on fatigue damage mechanisms. The most critical observation is that the Basquin intercept (log b) remains virtually constant, ranging from 3.67 to 3.70 across the 125 °C temperature interval (−40 °C to +85 °C). This consistency strongly indicates that the fundamental fatigue damage mechanism of the material—dominated by fiber breakage in unidirectional laminates—remains unchanged throughout this temperature range.

In contrast, the decrease in the Basquin slope (m) from 11.97 (RT) to 9.98 (HT), corresponding to a steeper S–N curve, indicates a faster rate of damage accumulation at elevated temperatures, particularly at high stress levels. While aligned with the minor matrix softening discussed in Section 4.2, this phenomenon is more precisely explained by the shear-lag theory. The reduction in matrix shear stiffness at 85 °C increases the ineffective length around broken fibers. This extended stress recovery zone generates larger stress concentrations on neighboring fibers, thereby accelerating the growth of fiber fracture clusters. This mechanism is significantly more pronounced under high-stress amplitude conditions where fiber breaks are frequent (low-cycle regime), compared to the high-cycle regime. Consequently, this selective acceleration of damage at high loads leads to the observed steepening of the S-N curve.4.4. Implications for Hydrogen Pressure Vessel Certification.

All findings of this study converge on the final application: hydrogen pressure vessel certification. International standards require stable and predictable performance across a demanding service temperature range (−40 °C to +85 °C).

Using a methodology free from testing artifacts, this study is the first to demonstrate that the carbon-fiber-reinforced polymer composites (Towpreg) quantitatively satisfy these certification requirements. The material (a) maintains over 93% of its static strength throughout the service temperature range, (b) exhibits a consistent fatigue damage mechanism (constant log b), and (c) displays minimal and predictable variations in fatigue sensitivity (m).

These results quantitatively confirm that the carbon-fiber-reinforced polymer composite (Towpreg) is a highly robust and reliable candidate material, offering long-term durability, fatigue resistance, and thermal stability required for next-generation Type IV hydrogen pressure vessels [6].

## 5. Conclusions

This study comprehensively evaluated the mechanical behavior of the carbon-fiber-reinforced polymer composite (Towpreg)—an emerging material for next-generation Type IV hydrogen pressure vessels—at three representative temperatures (−40 °C, 25 °C, and 85 °C) specified in hydrogen-tank certification standards. The key conclusions are as follows:(1)Methodological Contribution: The study identified the fundamental limitations of conventional large-chamber temperature testing (tab slippage at high temperature and energy inefficiency at low temperature) and introduced a new specimen-based localized thermal control methodology to overcome these issues.(2)High-Temperature System: The HY-Mini Heater System developed for 85 °C testing effectively prevented tab slippage and enabled reliable measurement of intrinsic material properties. Three-point thermocouple validation confirmed uniform temperature stabilization (±1.5 °C) within 20 min.(3)Low-Temperature System: The HY-Cooler System designed for −40 °C testing combined a Stirling cooler (utilizing the Joule–Thomson effect), a thermosiphon, and dual insulation (aerogel + foil) to achieve stable local cooling within 60 min using a compact and cost-efficient setup.(4)Material Process: A reproducible Towpreg plate fabrication process was developed by optimizing dry-winding and hot-press parameters according to the DM resin TDS, ensuring consistent specimen quality.(5)Performance Verification (Static): Static mechanical tests confirmed the excellent thermo-mechanical stability of the carbon-fiber-reinforced polymer composites (Towpreg), with tensile strength limited to approximately −7% at 85 °C and −2% at −40 °C relative to room temperature.(6)Performance Verification (Fatigue): Fatigue analysis revealed a nearly constant Basquin intercept (log b = 3.67–3.70) and a stable slope (m = 9.98–11.91) across all temperature conditions, demonstrating quantitatively that temperature variation has only a minor effect on the material’s strength and fatigue life.

In summary, this study (1) established a reproducible fabrication process for carbon-fiber-reinforced polymer composites (Towpreg) specimens, (2) developed innovative and reliable localized thermal testing systems, and (3) demonstrated that the carbon-fiber-reinforced polymer composites (Towpreg) maintain exceptional mechanical strength and fatigue durability across the demanding operating temperature range of hydrogen-storage systems. These findings confirm that the carbon-fiber-reinforced polymer composites (Towpreg) possess the fatigue life and thermo-mechanical stability required for practical implementation in next-generation hydrogen pressure vessels.

## Figures and Tables

**Figure 1 polymers-17-03241-f001:**
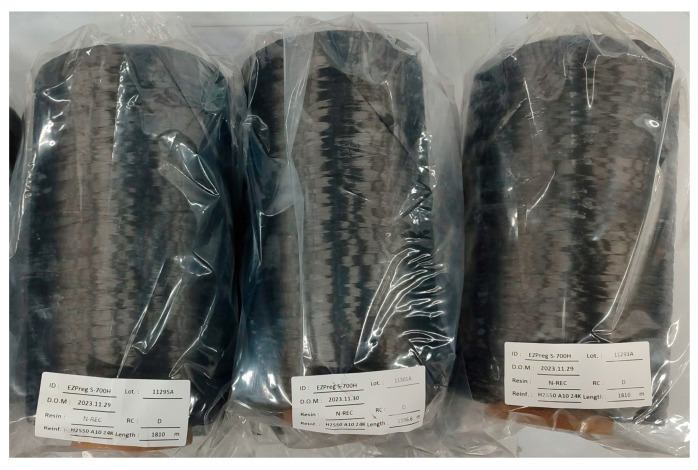
Towpreg (Hyosung H2550-24K/CeTePox).

**Figure 2 polymers-17-03241-f002:**
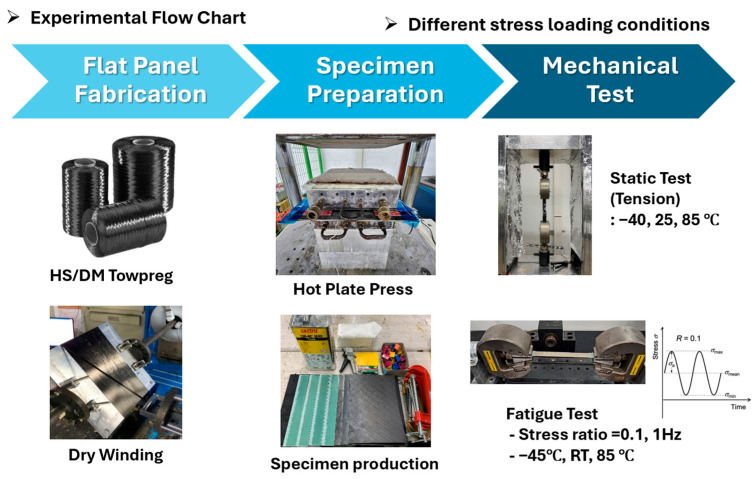
Experimental flow chart.

**Figure 3 polymers-17-03241-f003:**
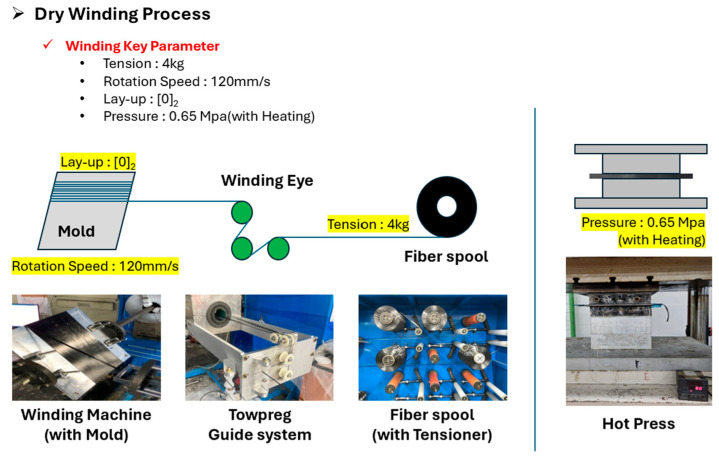
Dry winding process.

**Figure 4 polymers-17-03241-f004:**
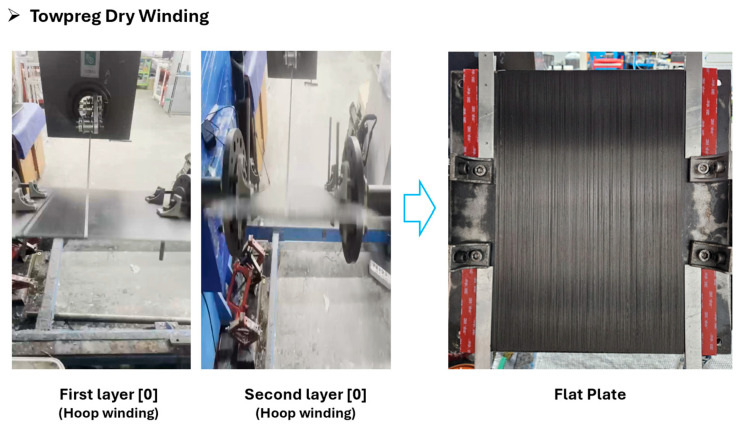
Towpreg dry winding process.

**Figure 5 polymers-17-03241-f005:**
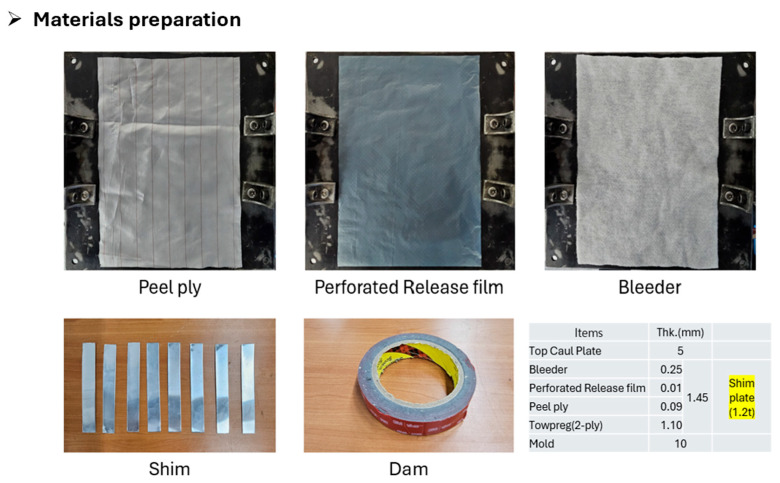
Material preparation for flat panel.

**Figure 6 polymers-17-03241-f006:**
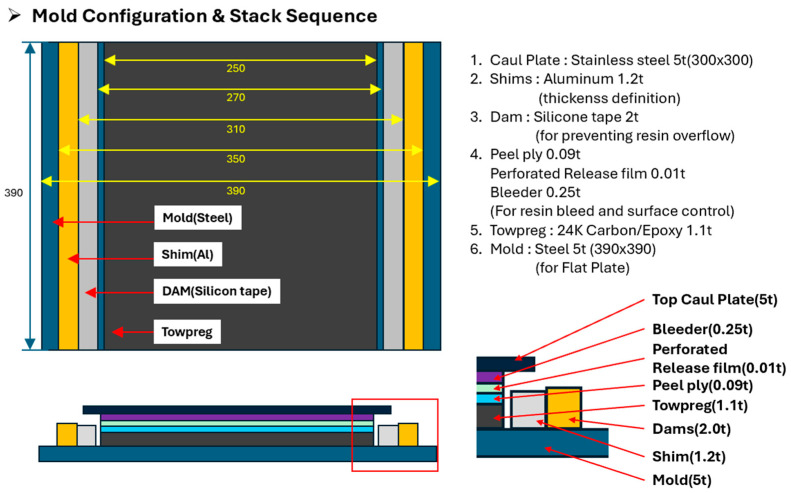
Mold configuration and stack sequence.

**Figure 7 polymers-17-03241-f007:**
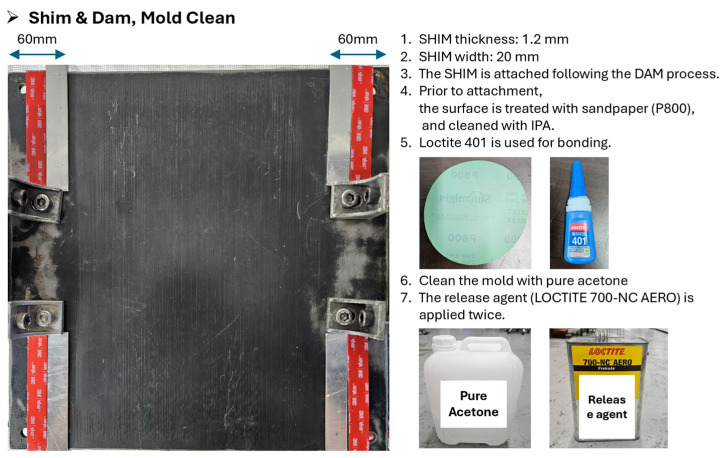
Shim & dam; Mold clean.

**Figure 8 polymers-17-03241-f008:**
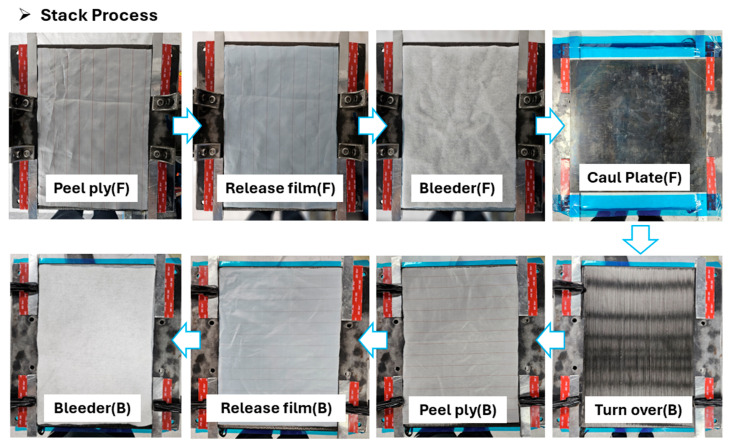
Stack process.

**Figure 9 polymers-17-03241-f009:**
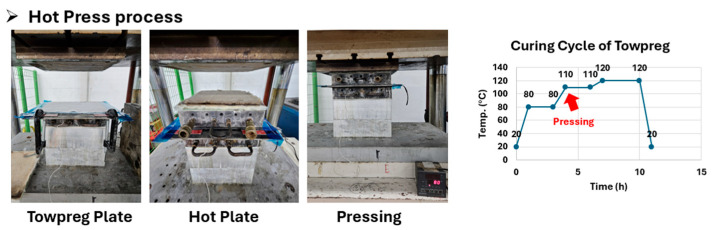
Hot press process.

**Figure 10 polymers-17-03241-f010:**
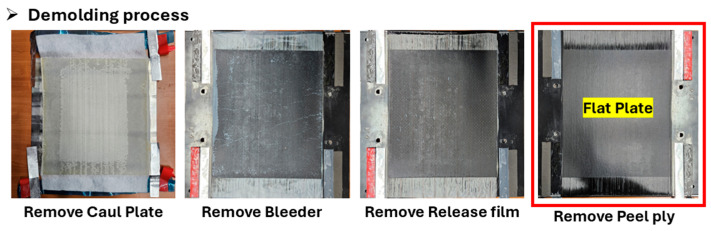
Demolding process.

**Figure 11 polymers-17-03241-f011:**
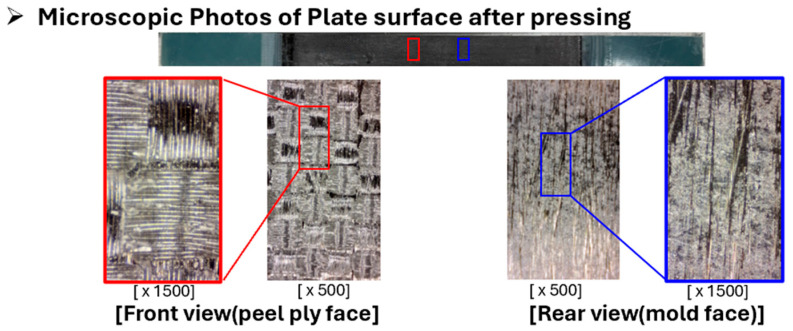
Microscopic photos of plate surface after pressing.

**Figure 12 polymers-17-03241-f012:**
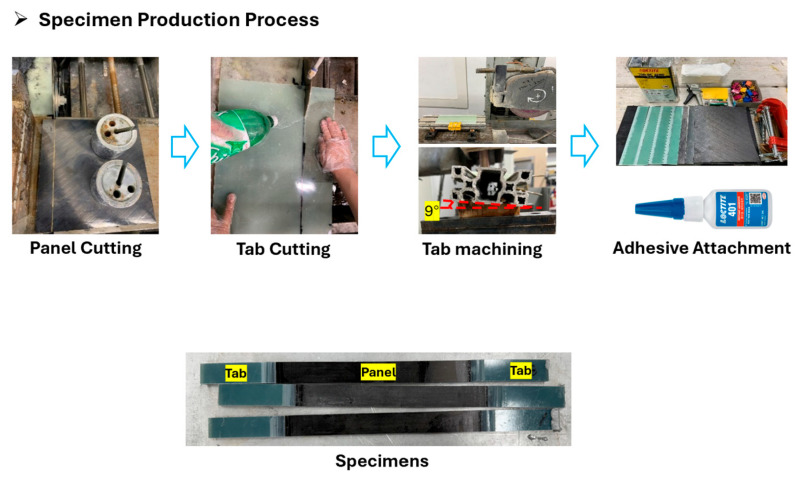
Specimen production process.

**Figure 13 polymers-17-03241-f013:**
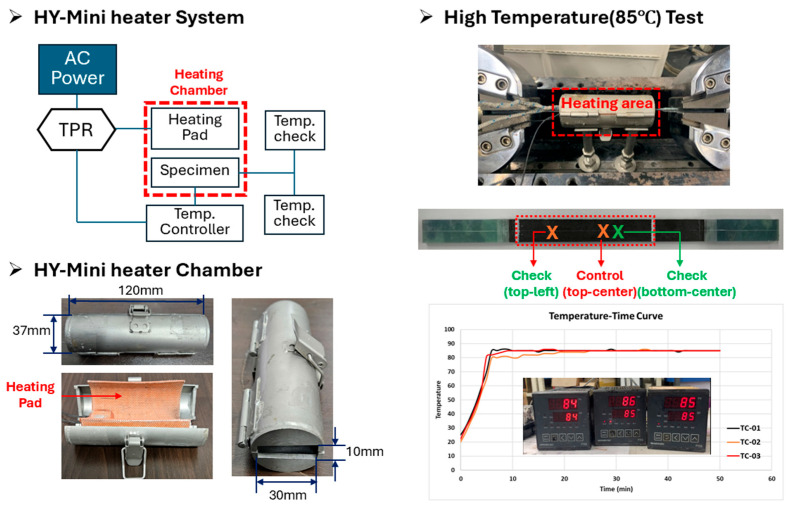
Test methods for high temperature (85 °C).

**Figure 14 polymers-17-03241-f014:**
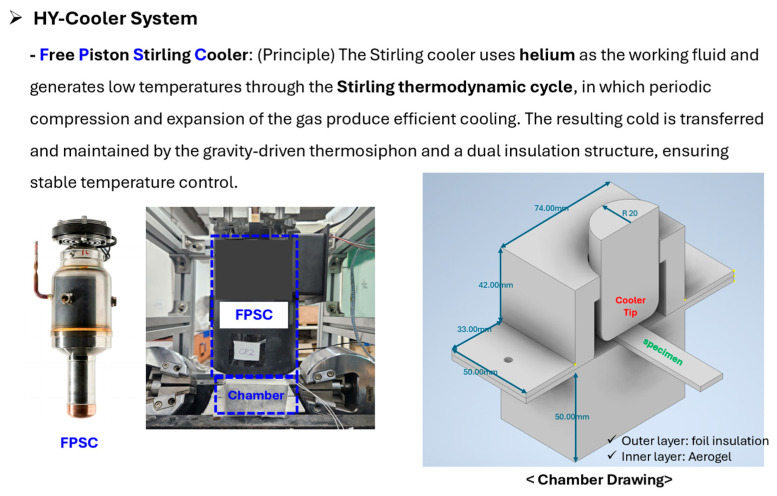
HY-Cooler System.

**Figure 15 polymers-17-03241-f015:**
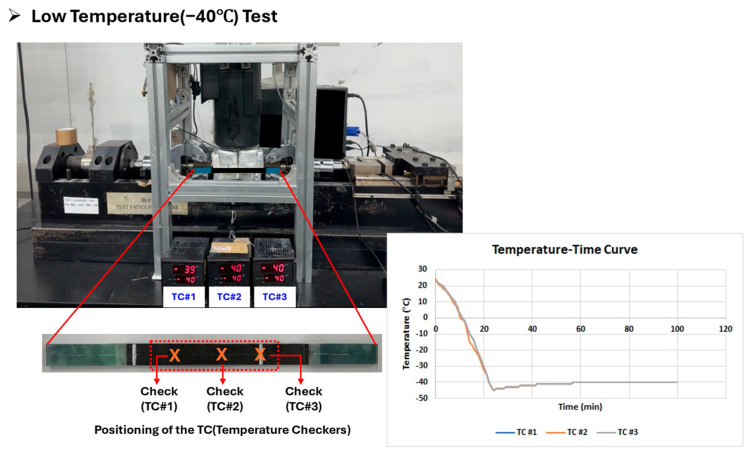
Test methods for low temperature (−40 °C).

**Figure 16 polymers-17-03241-f016:**
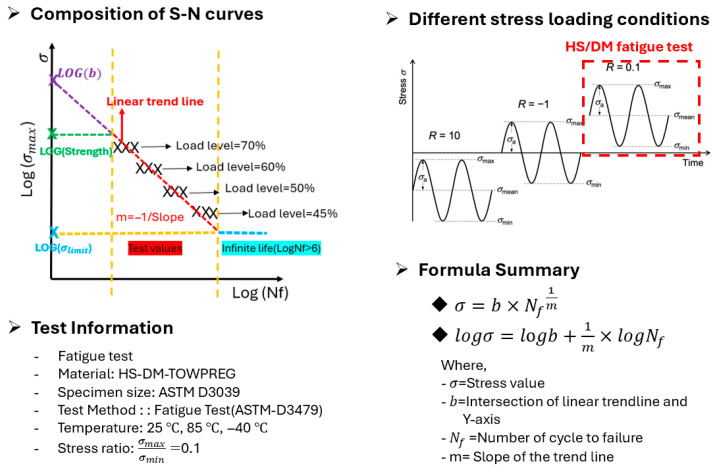
Theoretical background of fatigue test. Where σ = maximum stress, N_f_ = number of cycles to failure, m = slope, and log b = intercept [32].

**Figure 17 polymers-17-03241-f017:**
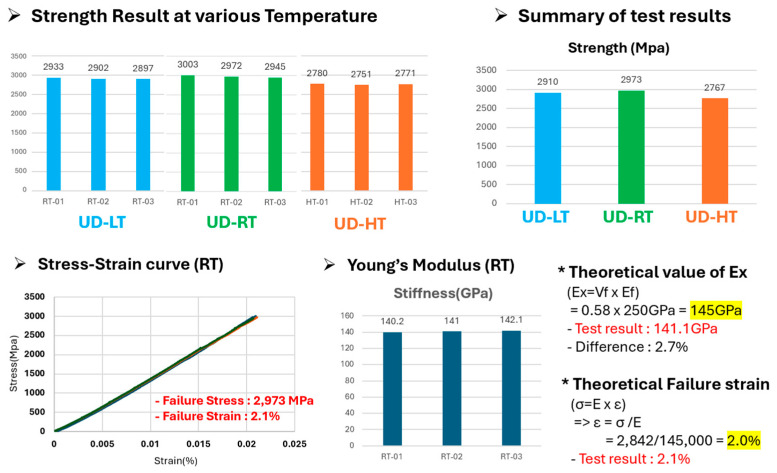
Tensile strength test results under various temperature.

**Figure 18 polymers-17-03241-f018:**
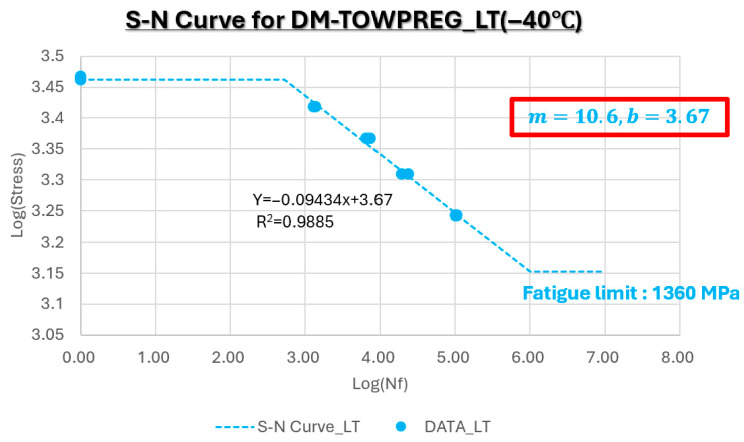
S–N curve for DM-Towpreg at −40 °C.

**Figure 19 polymers-17-03241-f019:**
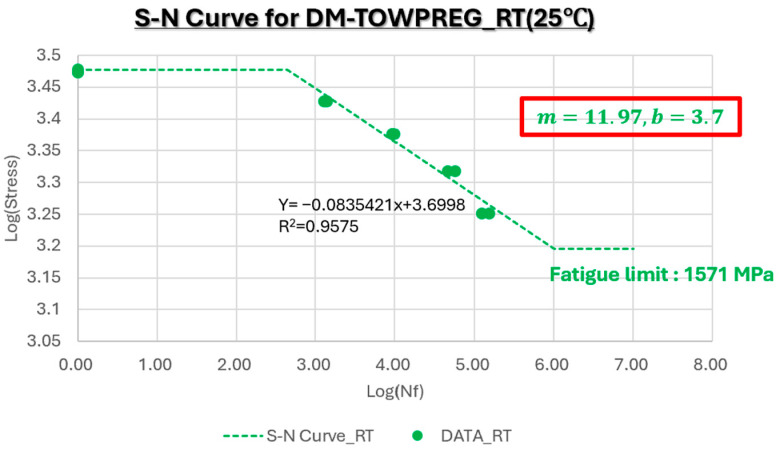
S–N curve for DM-Towpreg at 25 °C.

**Figure 20 polymers-17-03241-f020:**
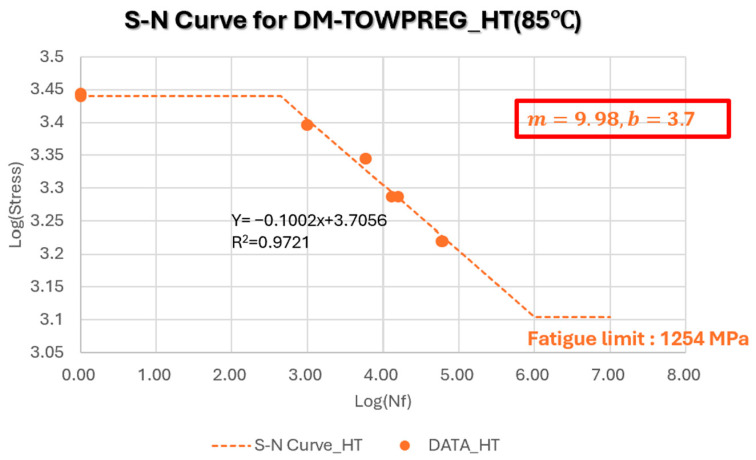
S–N curve for DM-Towpreg at 85 °C.

**Figure 21 polymers-17-03241-f021:**
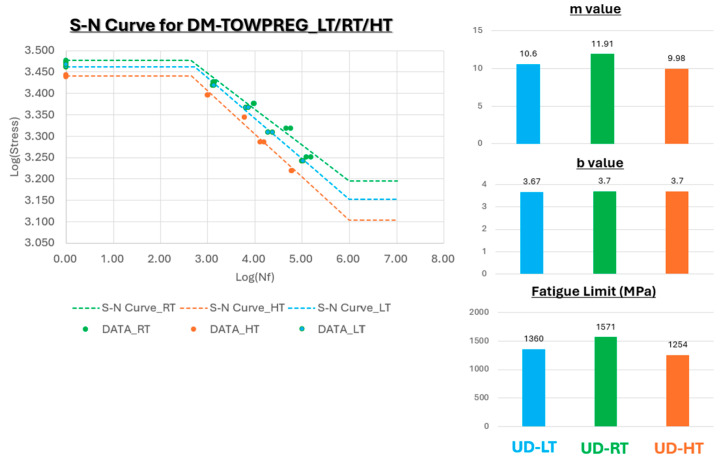
Fatigue test result under various temperature.

**Table 1 polymers-17-03241-t001:** Towpreg material properties.

Category	Property	Value
Reinforcement (Hyosung H2550-24K Carbon Fiber)	Filament Diameter (µm)	7.0
	Tensile Strength (MPa)	4900
	Tensile Modulus (GPa)	250
	Tensile Strain (%)	2.0
	Fiber Density (g/cm^3^)	1.78

**Table 2 polymers-17-03241-t002:** Matrix material properties.

Category	Property	Value
Matrix (CeTePox DM Epoxy System)	Glass Transition Temp, Tg (DSC) (°C)	127
	Initial Mixing Viscosity (@ 50 °C) (mPas)	590–600
	Gel Time (@ 110 °C) (min)	20–25
	Recommended Cure Condition	110 °C (1 h) + 120 °C (2 h)

**Table 3 polymers-17-03241-t003:** Optimized panel fabrication process parameters.

Process	Parameter	Value
Dry Winding	Winding Tension (kg)	4
	Winding Speed (mm/s)	Max. 120
Hot Press Curing	Pressure (MPa)	0.65
	Stage 1 (Temp./Time)	110 °C/1 h
	Stage 2 (Temp./Time)	120 °C/2 h
Mold Configure	Shim Thickness (mm)	1.2
	Dam Thickness (mm)	2.0

**Table 4 polymers-17-03241-t004:** Summary of static tensile properties by temperature.

Temperature Condition	Tensile Strength (MPa)	Strength (vs. RT)	Young’s Modulus (GPa)	Failure Strain (%)
−40 °C	2907.7	97.8%	-	-
25 °C	2973.3	100%	141.1	2.1
85 °C	2767.3	93.0%	-	-

**Table 5 polymers-17-03241-t005:** Summary of fatigue behavior and Basquin parameters by temperature.

Temperature Condition	Basquin Slope (m)	Basquin Intercept (log b)	Fatigue Limit (MPa)	Fatigue Limit (vs. RT%)
−40 °C	10.60	3.67	1360	86.6%
25 °C	11.97	3.70	1571	100%
85 °C	9.98	3.70	1254	79.8%

## Data Availability

Data is contained within the article.

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
