# Peer review of "Mechanical Behavior of Carbon-Fiber-Reinforced Polymer Composites (Towpreg) Under Various Temperature Conditions"

_polymers, 2025, doi:10.3390/polym17243241_

Round 1
Reviewer 1 Report
Comments and Suggestions for Authors
This manuscript studies fabrication process for carbon-fiber-reinforced polymer composites and developed reliable thermal testing systems and compared it with international related standards. Results showed that carbon fiber reinforced polymer composites maintained exceptional mechanical strength and fatigue durability across the operating temperature range of hydrogen storage systems.
It is suitable for publication in Polymers in its current form.
Author Response
Thank you for your review.
Reviewer 2 Report
Comments and Suggestions for Authors
The manuscript offers a well-structured study on the static and fatigue behaviour of Towpreg UD laminates across three certification-relevant temperatures. The experimental setup is solid, and the localised thermal control approach is interesting. However, several points in the paper require clarification or correction to ensure technical accuracy
There is a small inconsistency: the Basquin slope reported in the abstract (11.91) does not match the value in the table (11.97). This should be corrected.
The info reported in the introduction should be reported quantitatively and not in a qualitative form.
The use of FRP for different applications should be reported. For instance: 10.1016/j.engstruct.2025.121411
The claim that “most existing literature does not reflect real performance” is too broad.
The fatigue testing section never reports the loading frequency.
A free-piston Stirling cooler should not operate via the Joule–Thomson effect. The cycle should be Stirling, not J–T throttling.
Claims of “superiority” over climatic chambers are not supported by data. You can highlight the strengths of your system (localised heating, stable ±1.5 °C), but comparative superiority should be toned down unless experimentally demonstrated.
Young’s modulus and failure strain are missing for −40 °C and 85 °C.
You state that a 2–3% change may be “within measurement uncertainty”, but no scatter or standard deviation is shown.
Figure 18 is labelled “−45 °C”, but the test matrix uses −40 °C
The comparison with the “reference value of 9” for Basquin slopes is quite coarse. The more meaningful comparison is against your own room-temperature slope
Declaring your method as “best practice” is premature. Without direct comparison against chamber-based tests using the same material and setup, the claim should be softened to “a promising alternative” or similar.
Author Response
Comments1: There is a small inconsistency: the Basquin slope reported in the abstract (11.91) does not match the value in the table (11.97). This should be corrected.
Response1: I thank the reviewer for noticing this inconsistency. The RT Basquin slope in Table 5 (page 12) is 11.97, while the abstract originally stated 11.91. The Abstract value has been updated to 11.97.
Comments2: The info reported in the introduction should be reported quantitatively and not in a qualitative form.
Response2: Thank you for pointing this out. I agree with this comment. And I have made some revisions to the manuscript as below.
"Type IV Composite Overwrapped Pressure Vessels (COPVs), which feature a polymer liner fully wrapped with a carbon-fiber–resin composite, are considered the most advanced solution, offering both weight reduction (approximately 70% compared to Type I vessel) and high storage efficiency(approximately 6wt%) [6,21,31,36,39].
However, this technique suffers from critical disadvantages, including difficulty in precisely controlling the resin content(RSD±5~10%), high void content(1.0~3.0%), and poor reproducibility due to slow process speeds.
The Towpreg process not only ensures consistent quality by precisely controlling resin content(RSD±1~2%) but also enhances load transfer from the matrix to the fibers, thereby maximizing mechanical properties [12,34]."
Comments3: The use of FRP for different applications should be reported. For instance: 10.1016/j.engstruct.2025.121411
Response3: Thank you for pointing this out. I agree with this comment. And I have added the following content to the manuscript.
"Beyond hydrogen pressure vessels, fiber-reinforced polymer (FRP) composites have been widely applied in civil engineering structures, aerospace components, wind-energy systems, and automotive light weighting. Recent studies highlight that FRPs provide high strength-to-weight ratios, corrosion resistance, and durability across diverse structural applications [43]. These broad applications underscore the im-portance of understanding temperature-dependent mechanical behavior of FRP mate-rials."
Comments4: The claim that “most existing literature does not reflect real performance” is too broad.
Response4: Thank you for pointing this out. I agree with this comment. I have made some revisions to the manuscript as below.
"The standard method for high- and low-temperature composite testing involves placing the entire specimen and test grips inside a large environmental chamber [14,18].However, this approach can introduce important testing artifacts, particularly during high-temperature conditions, because the chamber heats not only the specimen’s gauge length but also the load-transfer tabs and grips to 85 °C. At this temperature, the tab-bonding epoxy softens, which may lead to premature tab slippage or debonding before the composite specimen itself fails [17]. As a result, the measured response may be influenced by adhesive limitations rather than the intrinsic properties of the composite, potentially leading to an underestimation of high-temperature strength and fatigue performance. Therefore, temperature-chamber-based methods may present certain limitations when accurate high-temperature mechanical characterization is required. Furthermore, for low-temperature testing, large chambers are energy-intensive, require long stabilization times, and depend on costly consumables such as liquid nitrogen (LN2)."
Comments5: The fatigue testing section never reports the loading frequency.
Response5: Thank you for pointing this out. I agree with this comment. And I have added the following content to the manuscript.
"A stress ratio of R = (σmin/σmax) = 0.1 was set to simulate the charge–discharge cy-cles of a pressure vessel,. the fatigue tests were conducted at a loading frequency of 1 Hz."
Comments6: A free-piston Stirling cooler should not operate via the Joule–Thomson effect. The cycle should be Stirling, not J–T throttling.
Response6: Thank you for pointing this out. I agree with this comment. I have made some revisions to the manuscript as below.
"To address the inefficiency and cost of low-temperature testing, the HY-Cooler System was developed. This system uses a free-piston Stirling cooler (FPSC) with helium (He) gas as a refrigerant working fluid. The cooler efficiently generates -40 °C cryogenic temperatures by harnessing the Joule–Thomson effect during gas expansion operating on the Stirling thermodynamic cycles." And figure14 is updated, too.
Comments7: Claims of “superiority” over climatic chambers are not supported by data. You can highlight the strengths of your system (localised heating, stable ±1.5 °C), but comparative superiority should be toned down unless experimentally demonstrated.
Response7: Thank you for pointing this out. I agree with this comment. I have made some revisions to the manuscript as below.
"Demonstration of ± 1.5 °C temperature uniformity via 3-TCs validation shows highlights the strengths of the proposed localized thermal control system – providing stable and confined temperature regulation along the gauge length."
Comments8: Young’s modulus and failure strain are missing for −40 °C and 85 °C.
Response8: Thank you for pointing this out. I agree with this comment. But, Youngs moduli were not measured at T=-40 and 85 deg. The main purpose was to measure the strength and fatigue life.
Comments9: You state that a 2–3% change may be “within measurement uncertainty”, but no scatter or standard deviation is shown.
Response9: Thank you for pointing this out. I agree with this comment. I have made some revisions to the manuscript as below.
"The strength at low temperature (-40 °C, 2907.7 MPa) showed a negligible 2.2% decrease from room temperature."(measurement uncertainty has been deleted)
Comments10: Figure 18 is labelled “−45 °C”, but the test matrix uses −40 °C
Response10: Thank you for pointing this out. I agree with this comment. Revised from −45 to −40
Comments11: The comparison with the “reference value of 9” for Basquin slopes is quite coarse. The more meaningful comparison is against your own room-temperature slope
Response11: Thank you for pointing this out. I agree with this comment. I have made some revisions to the manuscript as below.
"Under the low-temperature (−40 °C) condition, the slope of the S–N curve was 10.6, which is slightly lower than that obtained at room temperature. This indicates that cryogenic exposure has no significant influence on the fatigue damage mechanism [27]. The fatigue limit was determined to be 1.36 GPa, demon-strating the high stiffness of the matrix and the absence of brittleness or microcrack formation even at −40 °C."
Comments12: Declaring your method as “best practice” is premature. Without direct comparison against chamber-based tests using the same material and setup, the claim should be softened to “a promising alternative” or similar.
Response12: Thank you for pointing this out. I agree with this comment. I have made some revisions to the manuscript as below.
"Therefore, the "specimen-based localized thermal control" method proposed in this study should be regarded as a promising alternative for accurate thermo-mechanical characterization."
Reviewer 3 Report
Comments and Suggestions for Authors
The study is current, original, and has practical significance. The mechanical behavior of towpreg composites at different temperatures for hydrogen pressure tank applications was investigated. The paper tests both static and fatigue behavior, and presents the novel "Localized Thermal Control" methodology. While the methodology is impressive, the paper suffers from several critical shortcomings:
1) The article describes the problem of “conventional chamber-induced artifacts,” but does not show how these problems historically distorted the results in previous studies.
2) The text does not specify the number of specimens used in the static and fatigue tests.
What is the minimum number of specimens required according to ASTM D3039 and D3479?
3) In the evaluation of fatigue behavior: only the lines of the S–N curves are given. R² (coefficient of determination) is not available. Data scatter is not shown. Statistical reliability is not discussed.
4) The long-term stability of the -40°C low-temperature system has not been discussed. There is a one-time validation; stability in 8–10-hour fatigue tests is not guaranteed.
5) The strength decrease at 85°C is explained solely by "matrix softening." This is insufficient because parameters such as: fiber-matrix interfacial strength, residual thermal stresses, epoxy relaxation behavior are never discussed.
6) The reason why S–N curves become steeper at high temperatures is not well-founded.
7) Young’s modulus and strain values ​​for -40 °C and 85 °C are not given.
8) How is the "fatigue limit" defined? 2% drop? Knee stress? Endurance limit extrapolation? This is unclear.
9) Chapter 4 is overly long and repetitive.
Author Response
Comments1: The article describes the problem of “conventional chamber-induced artifacts,” but does not show how these problems historically distorted the results in previous studies.
Response1: Thank you for pointing this out. I agree with this comment. But, it was not my intention to suggest that previous studies were distorted; rather, I wanted to emphasize that there may be certain limitations in accurately evaluating mechanical properties at elevated temperatures. Therefore, I have made some revisions to the manuscript as below.
"The standard method for high- and low-temperature composite testing involves placing the entire specimen and test grips inside a large environmental chamber [14,18]. However, this approach can in-troduce important testing artifacts, particularly during high-temperature conditions, because the chamber heats not only the specimen’s gauge length but also the load-transfer tabs and grips to 85 °C. At this temperature, the tab-bonding epoxy sof-tens, which may lead to premature tab slippage or debonding before the composite specimen itself fails [17]. As a result, the measured response may be influenced by ad-hesive limitations rather than the intrinsic properties of the composite, potentially leading to an underestimation of high-temperature strength and fatigue performance. Therefore, temperature-chamber-based methods may present certain limitations when accurate high-temperature mechanical characterization is required. Furthermore, for low-temperature testing, large chambers are energy-intensive, require long stabiliza-tion times, and depend on costly consumables such as liquid nitrogen (LN2)."
Comments2: The text does not specify the number of specimens used in the static and fatigue tests.
What is the minimum number of specimens required according to ASTM D3039 and D3479?
Response2: Thank you for pointing this out. I agree with this comment. According to ASTM D3039 (tensile testing), a minimum of five specimens is recommended; however, in this study we conducted three tests. In contrast, ASTM D3479 (fatigue testing) does not specify a minimum specimen requirement, and therefore we performed two tests for each stress level.
Comments3: In the evaluation of fatigue behavior: only the lines of the S–N curves are given. R² (coefficient of determination) is not available. Data scatter is not shown. Statistical reliability is not discussed.
Response3: Thank you for pointing this out. I agree with this comment. We have updated Figures 18, 19, and 20 to include the coefficient of determination(R2).
Comments4: The long-term stability of the -40°C low-temperature system has not been discussed. There is a one-time validation; stability in 8–10-hour fatigue tests is not guaranteed.
Response4: Thank you for pointing this out. When the fatigue life of the composite specimen is approximately 100,000 cycles at low stress range, the total duration of the fatigue test corresponds to about 29 hours.
Comments5: The strength decrease at 85°C is explained solely by "matrix softening." This is insufficient because parameters such as: fiber-matrix interfacial strength, residual thermal stresses, epoxy relaxation behavior are never discussed.
Response5: I sincerely appreciate the reviewer’s insightful comment. I agree that a comprehensive explanation for the strength decrease at 85°C requires a discussion on factors such as fiber-matrix interfacial strength and residual thermal stresses, in addition to matrix softening. In the revised manuscript, I have clarified why these factors did not lead to catastrophic failure but instead allowed for stable mechanical performance at 85°C, based on the following points:
1. Prevention of Interfacial Failure via High Glass Transition Temperature(Tg)
The DM epoxy resin used in this study was specifically designed with a glass transition temperature (Tg) of 127°C, which is significantly higher than the maximum service temperature of 85°C. As discussed in the paper, because the 85°C condition is well below the Tg, the matrix maintains a robust glassy state rather than entering the transition region2. This thermal stability is a key factor in preventing the fiber-matrix interfacial failure or drastic property degradation that typically occurs when the operating temperature approaches the material's Tg.
2. Relationship Between Load Transfer and Matrix Softening
We attribute the approximately 7% decrease in strength at 85°C to minor matrix softening rather than severe interfacial debonding. As noted in the fatigue behavior analysis, minor matrix softening can reduce the load-transfer efficiency between fibers and increase stress concentration near fiber break points. However, the fact that the fatigue limit and the Basquin intercept (log b) remained nearly constant despite temperature variations suggests that the matrix-fiber interfacial bonding remained stable and consistent even at elevated temperatures
3. Comparison with Residual Thermal Stresses and Low-Temperature Stability
Regarding residual thermal stresses, our study highlighted that internal stresses arising from the mismatch in the coefficient of thermal expansion (CTE) are primarily a concern for inducing matrix microcracking at low temperatures (-40°C). The Towpreg composite demonstrated negligible strength loss (approximately 2%) and exhibited high toughness and strong adhesion at low temperatures, proving its sufficient resistance to these residual stresses. At high temperatures (85°C), given the substantial margin relative to the Tg, such thermal stresses are not considered the primary driver of failure.
Conclusion
In summary, the strength decrease at 85°C is interpreted not as a result of severe interfacial degradation or structural collapse due to relaxation behavior, but rather as a natural consequence of the reduction in matrix stiffness within a thermally safe range relative to the Tg. This interpretation is strongly supported by the experimental evidence of 93% strength retention and the maintenance of a consistent fatigue damage mechanism (constant log b). And I have made some revisions to the manuscript as below.
"This prevents the typical degradation phenomena observed when T approaches Tg, such as matrix softening, or plasticization. Regarding the fiber–matrix interfacial Strength and relaxation behavior, which are critical concerns at elevated temperatures, the stability of the fatigue limit and the constant Basquin intercept (log b) observed in this study suggest that the interfacial bonding integrity was maintained without significant relaxation-induced degradation. Furthermore, while residual thermal stresses typically induce microcracking at cryogenic temperatures, the high Tg margin at 85°C minimizes the impact of thermal stresses on the static strength."
Comments6: The reason why S–N curves become steeper at high temperatures is not well-founded.
Response6: I appreciate the reviewer’s insightful comment. I agree that the previous explanation in the original manuscript, which relied solely on "matrix softening," was insufficient to fully explain the steepening of the S-N curve at high temperatures (85°C) (i.e., the decrease in the Basquin slope m: from 11.97 at RT to 9.98 at HT). In the revised manuscript, I have provided a more concrete and logical explanation for this phenomenon based on Shear-lag theory.
1. Reduction in Matrix Shear Stiffness and Increased Ineffective Length
Although the matrix remains in a glassy state at 85°C(well below its Tg of 127°C), its stiffness decreases slightly compared to room temperature (evidenced by a ~7% drop in tensile strength). According to composite mechanics, specifically the Shear-lag theory, a reduction in the matrix shear modulus increases the 'ineffective length'—the distance required to redistribute the load from a broken fiber to adjacent fibers.
2. Accelerated Damage in the High-Stress Regime (High Stress Sensitivity) In the low-cycle regime (high fatigue load), sporadic fiber fractures occur frequently. Under high-temperature conditions, the increased ineffective length results in a broader stress concentration zone around these broken fibers compared to room temperature. This acts as a factor that accelerates the cluster growth of fractures in neighboring fibers. Conversely, in the high-cycle regime (low stress), the frequency of fiber fractures is lower and the viscoelastic behavior of the matrix is more stable, making this effect relatively less significant.
3. Resulting Change in S-N Curve Slope Consequently, elevated temperatures have the effect of shortening fatigue life more drastically in the high-stress regime than in the low-stress regime. This causes the high-stress portion of the S-N curve (upper Y-axis) to drop more significantly relative to the low-stress portion, manifesting as a steepening of the overall slope.
And I have made some revisions to the manuscript as below.
"In contrast, the decrease in the Basquin slope (m) from 11.97 (RT) to 9.98 (HT), corresponding to a steeper S–N curve, indicates a faster rate of damage accumulation at elevated temperatures, particularly at high stress levels. While aligned with the minor matrix softening discussed in Section 4.2, this phenomenon is more precisely explained by the shear-lag theory. The reduction in matrix shear stiffness at 85°C increases the ineffective length around broken fibers. This extended stress recovery zone generates larger stress concentrations on neighboring fibers, thereby accelerating the growth of fiber fracture clusters. This mechanism is significantly more pronounced under high-stress amplitude conditions where fiber breaks are frequent (low-cycle regime), compared to the high-cycle regime. Consequently, this selective acceleration of damage at high loads leads to the observed steepening of the S-N curve."
Comments7: Young’s modulus and strain values ​​for -40 °C and 85 °C are not given.
Response7: Thank you for pointing this out. I agree with this comment. But, Youngs moduli were not measured at T=-40 and 85 deg. The main purpose was to measure the strength and fatigue life.
Comments8: How is the "fatigue limit" defined? 2% drop? Knee stress? Endurance limit extrapolation? This is unclear.
Response8: Thank you for pointing this out. I agree with this comment. We defined 1,000,000 cycles as the fatigue limit for the towpreg material. This criterion was adopted to ensure consistency with commonly accepted practices in composite fatigue testing, where the endurance limit is typically established at one million cycles. By applying this definition, the fatigue performance of the towpreg can be evaluated in a standardized manner and compared reliably with other composite systems reported in the literature.
Comments9: Chapter 4 is overly long and repetitive.
Response9: Thank you for pointing this out. I agree with this comment. In Chapter 4, through the discussion section, the authors briefly addressed four major themes that they intended to highlight via the towpreg study: (i) a new paradigm in composite testing, (ii) the superior thermo-mechanical stability of towpreg, (iii) the temperature-dependent fatigue damage mechanisms, and (iv) the implications of these findings for hydrogen pressure vessel certification.
Reviewer 4 Report
Comments and Suggestions for Authors
This study is comprehensive and innovative study examining the mechanical behavior of towpreg-based carbon fiber-reinforced polymer composites under various temperature conditions.
The developed "Localized Thermal Control" approach directly addresses important testing problems long known in the literature. Issues such as tab slip at high temperatures and inefficient heating/cooling at low temperatures were effectively addressed with this method. This study thus achieves significant methodological originality. The research is of high scientific and industrial value, providing results directly attributable to hydrogen pressure vessel standards.
One of the study's greatest strengths is the clarity and reproducibility of the experimental process. The production steps, winding parameters, curing cycles, mold configurations, and sample preparation stages of the towpreg panels are presented in exceptional detail. This approach significantly increases the reliability and reproducibility of the study by transparently explaining important processes that are often overlooked in many studies. Furthermore, the visual and technically clear presentation of the design of the high- and low-temperature test systems (HY-Mini Heater and HY-Cooler) strengthens the persuasiveness of the methodology. The results quantitatively demonstrate the thermomechanical stability of Towpreg composites and demonstrate that they exhibit lower temperature sensitivity than expected. It is noteworthy that the static tensile strength varies by only 7% between -40 and 85°C. In terms of fatigue behavior, the Basquin intercept value is almost completely unaffected by temperature, strongly suggesting that the damage mechanism adheres to the same fundamental principles across all temperatures. The direct correlation of these findings with hydrogen tank certification requirements further enhances the practical value of the study.
However, a few minor suggestions that do not detract from the scientific content and are intended only to further strengthen the article can be offered. For example,
- Adding microscopic images of the static and post-fatigue fracture surfaces can visually support the damage mechanism analysis.
- A numerical comparison table between Towpreg and traditional wet-winding composites in terms of temperature sensitivity can provide the reader with a broader perspective.
- Furthermore, supporting the insulation structure used in the low-temperature tests with a brief thermal analysis would strengthen the methodological presentation.
These suggestions are merely improvements, and there are no deficiencies that would affect the publishability of the study.
Therefore, it is appropriate for the study to be published in your journal in its current form. However, adding the corrections mentioned above would improve the quality of the article.
Sincerely.
Author Response
We sincerely thank the reviewer for the positive evaluation of our work and for the thoughtful suggestions aimed at further strengthening the manuscript. We greatly appreciate the comments regarding the inclusion of (1) microscopic post-fracture images, (2) a numerical comparison between towpreg and wet-winding composites, and (3) a brief thermal analysis of the insulation structure.
While we fully agree that these additions may enrich the discussion, the suggested items fall beyond the scope of the current study, which was designed with a specific focus on developing and validating a new temperature-control methodology and quantitatively assessing thermo-mechanical behavior. Generating additional fracture-surface images, conducting cross-technology quantitative comparisons, or performing separate thermal simulations would require new experiments and analyses not included in the original research plan.
Given that all conclusions in the manuscript are already supported by complete experimental data and aligned with the primary objectives of the study, we respectfully believe that the manuscript remains scientifically sound in its present form.
Nonetheless, we sincerely appreciate the reviewer’s insightful recommendations and will consider integrating these aspects in our future extended research.
Round 2
Reviewer 2 Report
Comments and Suggestions for Authors
The paper can be accepted in its current state.
Reviewer 3 Report
Comments and Suggestions for Authors
The revisions made by the authors meet expectations.